# META-RCNN: META LEARNING FOR FEW-SHOT OBJECT DETECTION

## ABSTRACT

Despite significant advances in object detection in recent years, training effective detectors in a small data regime remains an open challenge. Labelling training data for object detection is extremely expensive, and there is a need to develop techniques that can generalize well from small amounts of labelled data. We investigate this problem of few-shot object detection, where a detector has access to only limited amounts of annotated data. Based on the recently evolving meta-learning principle, we propose a novel meta-learning framework for object detection named "Meta-RCNN", which learns the ability to perform few-shot detection via meta-learning. Specifically, Meta-RCNN learns an object detector in an episodic learning paradigm on the (meta) training data. This learning scheme helps acquire a prior which enables Meta-RCNN to do few-shot detection on novel tasks. Built on top of the Faster RCNN model, in Meta-RCNN, both the Region Proposal Network (RPN) and the object classification branch are meta-learned. The meta-trained RPN learns to provide class-specific proposals, while the object classifier learns to do few-shot classification. The novel loss objectives and learning strategy of Meta-RCNN can be trained in an end-to-end manner. We demonstrate the effectiveness of Meta-RCNN in addressing few-shot detection on Pascal VOC dataset and achieve promising results.

## 1 INTRODUCTION

Object detection is the task of identifying various objects in a given image, and localizing them with a bounding box. It is a widely studied problem in computer vision, and following the success deep convolutional neural networks (DCNN) in image classification (Karpathy et al., 2014; Krizhevsky et al., 2012), recent years have witnessed remarkable progress made in object detection based on deep learning. A series of detection algorithms based on DCNNs have been proposed which achieve state-of-the-art results on public detection benchmark datasets (Gidaris & Komodakis, 2015; Girshick et al., 2014; Ren et al., 2015; Lin et al., 2017a;b; Liu et al., 2016; Redmon & Farhadi, 2016). However, all these methods are data hungry, and require large amounts of annotated data to learn an immense number of parameters. For object detection, annotating the data is every expensive (much more than image classification), as it requires not only identifying the categorical labels for every object in the image, but also providing accurate localization information through bounding box coordinates. Moreover, in some applications, such as medical research, it's often impossible to even collect sufficient data to annotate. This warrants a need for effective detectors that can generalize well from small amounts of annotated data. We refer to the problem of learning detectors from limited labeled data as *few-shot detection*. For example, in *one-shot* detection, only one image is available with objects of interest annotated, and a detector needs to train on just this image and generalize. When presented with such small amounts of annotated data, traditional detectors tend to suffer from overfitting. Inspired by the fact that humans can learn a new concepts from little annotated data, we aim to develop a new few-shot detection algorithm.

There have been several efforts exploring few-shot learning (Vinyals et al., 2016; Finn et al., 2017; Snell et al., 2017). Many of them follow the principle of meta learning. In meta learning, a set of tasks in a few-shot setting is simulated from a large corpus of annotated data, and the model is optimized to perform well over these few shot tasks. This trains the model to learn *how* to solve few-shot tasks. However, most existing efforts of meta learning are mainly focused on classification. Adapting few-shot classification algorithms directly for few-shot detection (e.g. by replacing the

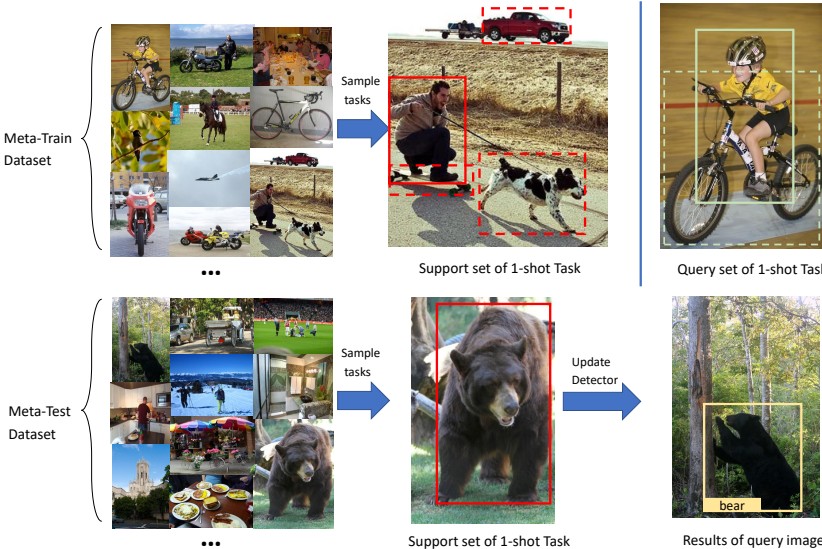

Figure 1: Few-Shot object detection in the meta-learning setting. From the meta-train dataset, a $K$ way-$N$ shot support set and a query set are sampled to create a task. The meta detector makes predictions on the query set by using the knowledge from the support set, and updates the detector based on the loss on the query set. In the example above, there are many objects (person, dog and truck), but it is annotated with the goal of detecting only a person. At test time, a single annotated image from a novel class (bear) is available for the detector to learn a model that can generalize.

region classification branch of detector with a meta-learner) is non-trivial because of two major concerns: i). Detection algorithms not only require classifying objects but also need to correctly localize objects in cluttered backgrounds by using a Region Proposal Network (RPN) and bounding box (bbox) regressors. It is thus also desirable that both RPN and bbox regressors should also be capable enough to adapt to few-shot settings. ii). For a given task with one (or few) annotated image(s), the annotated image may contain objects from several classes. But only a few objects of interest are annotated. The goal of the few-shot detector is to detect only these objects of interest. Unfortunately, a naively trained meta-detector's RPN would detect all objects (even objects from classes not of interest) and try to classify them as one of the classes of interest rather than background images (See Figure 1 for an example).

We aim to address these challenges by proposing a novel method for solving few-shot detection using the meta-learning paradigm. We develop Meta-RCNN, an end to end trainable meta object detector. The proposed Meta-RCNN follows the episodic learning paradigm of meta-learning (Vinyals et al., 2016), where based on a give meta-train dataset, multiple few-shot tasks are simulated. For a given task, we first construct a class prototype for each of the annotated object categories in the support set. Using these prototypes, a class-specific feature map of the entire image is constructed, i.e., we obtain a feature map of the entire image for each of the class prototypes. These feature maps are tailored to detect only objects of the class of the prototype, by giving higher attention to appropriate regions in the image containing that object. Finally, all feature maps are merged to produce a combined feature map, followed by an RPN, and then classification and bbox regression layers.

Meta-RCNN learns few-shot detector where the whole framework can be trained via meta-learning in an end-to-end manner. In contrast to the naive adaptation of meta-learning for classification into an object detection framework, Meta-RCNN learns the few-shot classifier, the RPN, and the bbox regressor in the meta-learning setting, thus making all three components suitable for handling few-shot scenarios. Moreover, Meta-RCNN learns a class-specific feature map for a given class prototype enabling easier distinction between classes of interest and backgrounds (where other objects in the image from classes not of interest are considered as backgrounds). We demonstrate the effectiveness of Meta-RCNN on two few-shot detection benchmarks: Pascal VOC and animal subset of ImageNet, and show that Meta-RCNN significantly improves the detection result in few shot settings.

## 2 RELATED WORK

**Generic Object Detection**. Object detection based on deep learning can be broadly divided into two families: two-stage detectors and one-stage detectors. Two-stage detectors such as RCNN (Girshick et al., 2014), Fast RCNN (Gidaris & Komodakis, 2015) and Faster RCNN (Ren et al., 2015), first generate a sparse set of proposal candidates, and a fixed-length feature vector is extracted from each of these candidates, followed by a categorical classifier and a bounding box regressor. Two-stage detection algorithms have achieved state-of-the-art results on many public benchmarks (He et al., 2016; Lin et al., 2017a), but are relatively slower than one-stage detectors. One-stage detectors such as SSD (Liu et al., 2016), Yolo (Redmon et al., 2016; Redmon & Farhadi, 2016) and RefineDet (Zhang et al., 2018) directly generate categorical proposals from the feature map and thus avoid cascaded region classifiers. One-stage detectors can achieve real-time inference speed but the detection accuracy is often inferior to two-stage detection algorithms. Both detection families assume access to a large set of annotated data, and are not suitable for scenarios where the model has access to small amounts of annotated training data. In contrast, our proposed Meta-RCNN method addresses detection problem of few-shot setting, and achieves promising results.

**Meta Learning for few-shot classification**. Few-shot learning has been widely explored in image classification, and currently the most promising methods are mainly based on meta learning. Ravi & Larochelle (2016) optimized the base-model via an LSTM-based meta-learner which simulates traditional SGD optimization method. Finn et al. (Finn et al., 2017) proposed MAML which learns a good feature initialization which can adapt to a new task in only one gradient step udpate. Based on MAML, Li et al. (2017) proposed Meta-SGD which learns a set of learnable parameters to control gradient step of different tasks. Learning initialization is potentially a very general idea for few-shot learning however, the training process can be unstable (Antoniou et al., 2018) especially for complex problems such as detection. Vinyals et al. (Vinyals et al., 2016; Snell et al., 2017) proposed a matching network which followed a non-parametric principle by learning a differentiable K-Nearest Neighbour model. Ren et al. (Ren et al., 2018) extended this idea to semi-supervised learning by self-learning from the unlabeled data. Sung et al. (Sung et al., 2018) proposed a relation network to automatically define the optimal distance metric. These metric-learning based methods are easy to train and effective in addressing few-shot classification. However, directly adapting these techniques for detection is very challenging as just replacing the object classification branch of a detector with a meta-learner is not sufficient, and training the RPN under a meta-learning paradigm is non-trivial.

**Few-shot Object Detection**. Few-shot detection has received considerably less interest from the community. Dong et al. (Dong et al., 2018) addressed few-shot detection using large scale unlabeled data. Their model is based on a semi-supervised method which extracts knowledge from unlabeled dataset to enrich training dataset by self-paced learning and multi-modal learning. However, their method may be misled by the incorrect predictions from initial model and also requires re-training the model for every new task. Chen et al. (Chen et al., 2018) propose a Low-shot Transfer Detector (LSTD) using regularization to transfer the knowledge from source domain to target domain by minimizing the gap between these two domains. RepMet (Schwartz et al., 2019) is a few-shot detection algorithm based on meta learning. It replaces the fully connected classification layer of a standard detector with modified prototypical network. However, they suffer from the two limitations of the RPN and bbox regression not being able to handle few-shot settings, and difficulties in distinguishing object classes of interest from background (including object classes not of interest). Our proposed method is also based on meta learning but can be optimized end-to-end and addresses these limitations to do effective few-shot detection.

## 3 PRELIMINARIES

### 3.1 PROBLEM SETTING

In this section we present the formal problem setting of few-shot detection investigated in our paper. Assume we have two datasets $L$ and $S$, where $L$ is a large scale annotated dataset with $L_c$ categories and $S$ is a dataset with only a few annotated images with $S_c$ categories. There is no category overlap between two datasets: $L_c \cap S_c = \phi$. Our goal is to learn a robust detector based on the annotated data in $L$ and $S$ to detect unlabeled objects of $S$.

The proposed Meta-RCNN aims to learn a general detection framework which can be quickly adapted to different detection tasks which have only a few labeled samples. We follow the standard training scheme of meta learning, which splits the whole learning stage into two parts: meta-training and meta-testing, and the model is optimized over multiple few-shot tasks simulated from the meta-training data. Specifically, during meta-training, few-shot detection tasks are sampled from $L$, and each task contains a support set and a query set. For the $i$-th task, $K$ ways (or categories) and $N$ images per category are randomly selected from $L_c$ to build support set: $T_i^{L,s}$. Similarly, $Q$ images per category are randomly selected to build query set $T_i^{L,q}$. Support set $T_i^{L,s}$ and query set $T_i^{L,q}$ construct a complete task extracted from $L$ (See Figure 1):

$$T_i^L = \left\{ T_i^{L,s}, T_i^{L,q} \right\} \tag{1}$$

where both the support set and query set are used to train the meta-model. The meta-model optimizes the base-model with respect to the support set and makes predictions on query set. Finally the loss suffered on the query set is used to update the model. In the meta-testing stage, similar to meta-training stage, a set of few-shot tasks are sampled from $S$:

$$T_i^S = \left\{ T_i^{S,s}, T_i^{S,q} \right\} \tag{2}$$

where $T_i^{S,s}$ is support set and $T_i^{S,q}$ is query set. The model makes predictions on the query set, and these results are averaged across several few-shot tasks to evaluate the expected performance of the few-shot detector over a variety of novel few-shot detection tasks.

## 3.2 Overview of Faster RCNN

Meta-RCNN is based on two-stage region based object detection algorithms. In this paper, we use the state-of-the-art detection algorithm Faster RCNN (Ren et al., 2015) as our base model, which is widely used in the computer vision community. Faster RCNN consists of two components, an RPN (Region Proposal Network) for proposal generation and Fast RCNN for region classification. RPN generates a sparse set of proposals which are classified into different categories by the region classifiers. Specifically, RPN extracts a feature vector from each region by scanning the whole image using sliding windows. This is followed by a binary classifier (objects vs backgrounds) and a bounding box regressor, where easy negatives are filtered. For each proposal, a fixed-length feature vector is extracted by using ROI Pooling layers. This vector is then fed into a sequence of dense connected layers branching into two outputs. One output is responsible for representing softmax probability over $K + 1$ classes($K$ target classes and one background class), and the other one encodes four real-values for refining bounding box position. We denoted $u$ and $v$ as the category and bounding box label respectively, $p$ as the predicted probability distribution over C classes, and $t_u$ as the predicted bounding box prediction of class $u$, and $\lambda$ as the trade-off parameter. $L_{\text{cls}}$ represents softmax loss and $L_{\text{loc}}$ represents SmoothL1 loss function. The entire network can be optimized in an end-to-end manner by minimizing loss $L(p, u, t^u, v)$:

$$L(p, u, t^u, v) = L_{\text{cls}}(p, u) + \lambda [u \geq 1] L_{\text{loc}}(t^u, v), \tag{3}$$

However, two-stage detectors require a lot of training samples to obtain a good performance. In the next section, we present the proposed Meta-RCNN which builds over Faster RCNN and is specifically designed to address few-shot detection.

## 4 Meta-RCNN

### 4.1 Overview

We now present our proposed method Meta-RCNN for few-shot detection (See Figure 2 for an overview). Meta-RCNN is trained with multiple few-shot tasks simulated from the meta-train dataset. For each episode, a few object categories of interest are assumed to be annotated (Support set). During meta-training, a prototype is computed for each object category. For each of these category prototypes, a class-specific feature map is generated by using a class-attention module which combines the prototype information with the feature map of the entire image. This feature map only highlights the signals of the class of interest, and suppresses information from other classes. Finally,

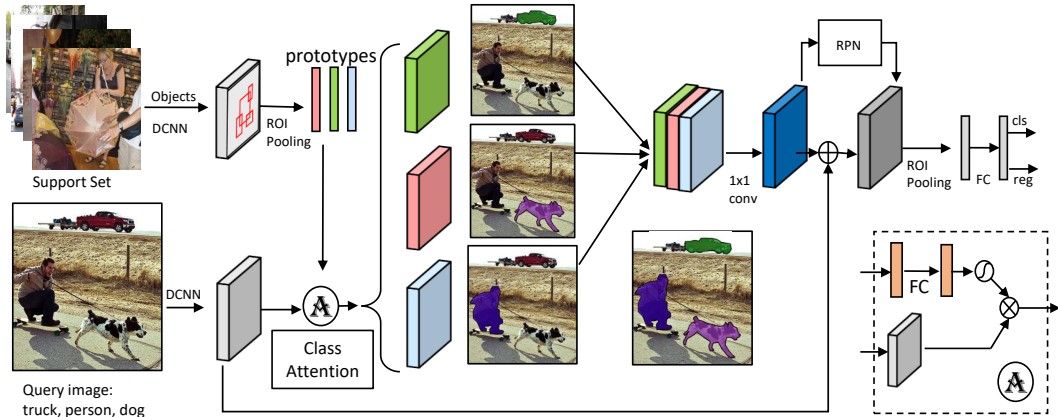

Figure 2: The Meta-RCNN workflow. A set of prototypes of different categories are extracted from the support set. For each class, conditioned on these prototypes, a class-specific feature map from query set is generated by applying the class attention module to the feature map of the entire image. The new class-specific feature map is tailored to detecting objects of that specific class. The class-specific feature maps are concatenated together and finally, an RPN is applied followed with region classification layer and bounding box regressors. The whole network is optimized via meta learning and can be trained end-to-end.

feature maps of all target categories are combined, followed by RPN and RCNN branches to make predictions on the query set. Based on the loss on the query set, the model is updated.

Meta-RCNN is general paradigm to train few-shot detector via by meta-learning. For each task, irrelevant categories and background can be filtered by attention module, and the final generated feature map learns a general representation for the given few-shot detection task. Compared to other variants (Schwartz et al., 2019) which directly replaces the FC classification branch with a meta-learning branch, Meta-RCNN is more general and the whole framework can be optimized including RPN and bbox regressors, making all the components few-shot capable. Next, we present the details of the model.

## 4.2 META-TRAINING

During Meta-Training, multiple $K$ way-$N$ shot tasks are simulated from the annotated dataset $L$. To fit memory size, in Meta-Training stage we train the model using 5way-1shot tasks, and only 5 query images (1 query image per class). This results in a total of 10 images for one task. With this, implementing the meta-training is not too difficult. For each task $T_i^L$, images of support set $T_i^{L,s}$ are fed into Faster RCNN to generate region features. For each of the object categories of interest (those assumed to be annotated in the support image), a prototype $P_c$ is generated based on the corresponding region features:

$$P_c = \frac{1}{N_c} \sum_i^{N_c} r_c^i \qquad (4)$$

where $P_c$ denotes prototype of class $c$, and $r_c^i$ denotes $i$-th region features of all annotated objects from class $c$. Based on these generated prototypes, images of query set $T_i^{L,q}$ are fed into the same Faster RCNN model and we obtain the image feature map before RPN and ROI Pooling. For each category, a class-specific feature map is learned based on the input query image and its corresponding prototype. We use a learnable class attention module here to highlight the signals of target class and suppress signals of other categories. The class attention module is based on basic channel-wise multiplication. The prototype $P_c$ is encoded by a FC layer $\phi$, which is later combined with feature map $f$ by element-wise multiplication:

$$F_c = f \odot \phi(P_c) \qquad (5)$$

For each category $c$, one new feature map $F_c$ is generated which aims to highlight the objects of class $c$. Next, all these new feature maps are combined into one feature map $F$:

$$F = \text{concate}\{F_1, F_2, ..., F_k\} \tag{6}$$

$F$ learns a general representation of K-classes, where each sub-channel contains information of different classes of interest. Based on the new feature map $F$, 1x1 conv layer is used to reduce computation cost, followed by RPN to produce region proposals. In order to recover the information lost in attention module, we finally combine the new generated feature map with original feature map by element-wise summation, and crop region features based on the new generated map. Finally, a K+1 region classifier and a bbox regressors are optimized w.r.t the label info from query set $T_i^{L,q}$:

$$L(T_i^{L,q}; T_i^{L,s}, \theta) = L_{\text{loc}} + L_{\text{cls}} + L_{\text{RPN}} \tag{7}$$

where $\theta$ represents the parameters of Meta-RCNN.

### 4.3 META-TESTING

During meta-testing, we sample few-shot detection tasks from $S$. The annotations of support set are available and we make predictions on the query set to evaluate the performance of Meta-RCNN. For each task $T_i^S$, prototypes are generated from support set $T_i^{S,q}$, which are later used to generate new class-specific feature maps of images from query set $T_i^{S,q}$. In this stage, we need to finetune the model based on the labeled images of support set. The finetuning operation addresses the learning limitation of non-parametric method when more labeled images are provided. Finally, we evaluate the output from the query set as traditional detection problem:

$$p, u = \text{MetaRCNN}(T_i^{S,q}; T_i^{S,s}, \theta) \tag{8}$$

where $p$ is class probability vector and $u$ is location set of bounding boxes.

## 5 EXPERIMENTS

### 5.1 DATASETS AND IMPLEMENTATION DETAILS

| DATASET | Train | #Img | #cls | Test | #Img | #cls |
|---------|-------|------|------|------|------|------|
| **VOC-FSOD** | VOC2007trainval | $\sim 4.9$k | 10 | VOC2007test | $\sim 2.2$k | 10 |
| **IMAGENET-FSOD** | ImageNet-LOC | $\sim 53$k | 100 | ImageNet-LOC | $\sim 117$k | 214 |

Table 1: Two few-shot object detection benchmark testbeds for performance evaluation

**Benchmark Datasets:** We construct two benchmark testbeds to facilitate the performance evaluation for few-shot object detection in meta-learning settings. The first is on Pascal VOC2007, and the second is on the animal subset of ImageNet-LOC dataset. Table 1 gives details of these datasets. Pascal VOC2007 has 20 categories with 5k images in trainval set and 5k images in test set. A subset of 10 categories are randomly from selected for VOC2007 trainval set for Meta-Training and the remaining 10-category subset of VOC2007 test set is used for Meta-Testing. Images without target object categories are removed. For ImageNet-FSOD benchmark, we use the subset of first 100 animal classes of ImageNet in Meta-Training stage and the subset of remaining 214 animal species in ImageNet-LOC in Meta-Testing stage. The model used in VOC-FSOD benchmark is pre-trained on ImageNet, while in ImageNet-FSOD benchmark, the model is pre-trained on MSCOCO dataset with 115k images in 80 categories.

**Task Generation:** For each benchmark, Meta-RCNN is evaluated on multiple tasks with different $K$way-$N$shot few-shot settings ($N$ annotated images per category). For VOC-FSOD benchmark, we have 3 few-shot settings to evaluate Meta-RCNN: 5way-1shot, 5way-3shot and 5way-5shot. In detection, a single image has more than one object, and proposal generation will automatically increase the number of training samples, so the real number of training samples is about 5 times larger than $N$. On ImageNet-FSOD benchmark, we mainly follow (Chen et al., 2018) and (Schwartz et al., 2019) with two settings: 50way-1shot and 50way-5shot.

**Meta-model Parameter Setting:** In Meta-Training stage, totally 1000 distinct tasks and 5000 tasks are generated in VOC-FSOD benchmark and ImageNet-FSOD benchmark respectively. There are 10 images per class in query set to update the model weights for 10 epochs. The initial learning rate is set to 1e-3 and is reduced to 1e-4 every 600 tasks and 3500 tasks in VOC-FSOD benchmark and ImageNet-FSOD benchmark. We set the batch size as 5 during query update.

**Basic Detection Parameter Setting:** The parameter settings in Meta-RCNN is identical to vanilla Faster RCNN. Proposal overlap with objects larger than 0.5 are considered positive and less than 0.3 are negative. During Meta-Training the top 128 confident proposals are selected for training and during evaluation, 300 proposals with largest confidence score are selected. we build our Meta-RCNN based on Faster RCNN with VGG16 (Simonyan & Zisserman, 2014) and ResNet50 (He et al., 2016) model which is pretrained on ImageNet.

**Model Evaluation:** We evaluate Meta-RCNN based on multiple tasks of few-shot settings, which follows the evaluation metric of standard meta learning. More specifically, in evaluation stage, 200 $K$shot-$N$shot tasks are sampled from dataset $S$ and images in query set will be evaluated. Mean average precision(mAP) over selected $K$ categories is used as evaluation metric.

## 5.2 RESULTS ON VOC-FSOD BENCHMARK

We validate the effectiveness of Meta-RCNN on VOC-FSOD benchmark where subset of 10 VOC categories are selected for Meta-Training and the other ten categories are used for Meta-Testing. For fair comparison, these two subsets are split as similar as possible. For example, we keep animal categories on both sides since they share similar semantic information (see appendix for details). Here we set up three baselines on VOC-FSOD benchmark to compete with proposed Meta-RCNN.

- **vanilla FRCN** (Ren et al., 2015): the vanilla Faster RCNN which is the most popular object detection algorithm with competitive performance on many benchmarks. The vanilla FRCN is not designed for few-shot detection problem, but we try to include this baseline by fine-tuning the detector on the few-shot training data.

- **LSTD** (Chen et al., 2018) is a few-shot detection algorithm based on Faster RCNN. LSTD uses categorical regularization items which transfers knowledge of $L$ dataset to $S$ dataset.

- **FRCN-PN** is a modified version of Faster RCNN using meta-learning, which replaces final FC classification layer with non-parametric prototypical network (PN), sharing the same principle of RepMet (Schwartz et al., 2019).

All three baselines as well as the proposed Meta-RCNN are based on VGG16 (Simonyan & Zisserman, 2014) backbone. For Regular FRCN and LSTD, we first train a global Faster RCNN during Meta-Training. Then the pretrained detector models are adapted to different tasks during Meta-Testing. During Meta-Testing, Meta-RCNN and vanilla FRCN are finetuned for 4 epochs while LSTD requires longer finetuning period (10 epochs). For FRCN-PN, prototypes of different categories are extracted as Meta-RCNN, and metric distances are learned to assign correct labels to each proposal. We report the results on Table 2 based on three different settings.

From Table 2, the performances of all four methods improve with training shot increasing. Notably, FRCN-PN obtains much less improvement because the non-parametric property of PN layer limits its learning capacity from increased training samples. Benefit from the finetuning operation as well as FC layer in final classification and regression, Meta-RCNN can still maintain consistent improvement when trained with more samples. Furthermore, it's interesting that Regular FRCN outperforms FRCN-PN even in very few-shot cases (5way-1shot), where non-parametric property does not help PN obtain better performance. We argue this is because few-shot detection problem is more difficult than few-shot classification problem, as we discussed in introduction section. FRCN-PN cannot learn a representative prototype of background classes and the whole framework cannot be optimized by meta learning style (e.g., RPN and bbox regressors). The failure of FRCN-PN indicates naively attach components from few-shot classification framework cannot address few-shot detection problem. Finally, our Meta-RCNN achieves better results than all three baselines.

**Performance of RPN:** Here, we present the performance of RPN to validate our concerns of the negative impact of irrelevant categories. We use regular FRCN and FRCN-PN as our baseline. The models are optimized in the same manner as before but during Meta-Testing, we evaluate the recall on each task instead of mAP. From Table 3, Regular FRCN baseline outperform the FRCN-PN significantly. This is because objects of irrelevant categories in the same image hurt the training process of RPN. And our proposed Meta-RCNN outperforms these two baseline significantly. Meta-

| Method | 5way-1shot | 5way-3shot | 5way-5shot |
|---|---|---|---|
| vanilla FRCN (Ren et al., 2015) | 14.78% ± 1.02% | 20.34% ± 1.26% | 26.89% ± 1.23% |
| LSTD (Chen et al., 2018) | 17.66% ± 1.65% | 22.37% ± 0.81% | 29.00% ± 1.28% |
| FRCN-PN | 12.71% ± 0.70% | 13.91% ± 0.70% | 14.33% ± 0.61% |
| Meta-RCNN (ours) | **19.22% ± 1.01%** | **24.45% ± 1.20%** | **31.11% ± 0.88%** |

Table 2: mAP Performance Evaluation on the VOC-FSOD BENCHMARK

RCNN learns a general feature map for all $K$ way-$N$ shot detection problem and optimize RPN by meta learning scheme, which proves more effective in few-shot settings. Notably, the results are surprising since the recall of RPN in few-shot scenario is significantly lower ($> 90\%$ with enough training data on VOC dataset).

| Model | Backbone | 5way-1shot | 5way-3shot | 5way-5shot |
|---|---|---|---|---|
| vanilla FRCN (Ren et al., 2015) | VGG16 | 24.9% | 26.5% | 28.4% |
| FRCN-PN | VGG16 | 24.7% | 24.9% | 26.1% |
| Meta-RCNN (ours) | VGG16 | **26.1%** | **27.9%** | **33.7%** |

Table 3: Recall evaluation of Meta-RCNN on VOC-FSOD BENCHMARK test set.

### 5.3 RESULTS ON IMAGENET-FSOD BENCHMARK

On ImageNet-FSOD benchmark, we adapt weights of detector pretrained on MSCOCO trainval set, and then optimize Meta-RCNN based on this starting point. The Meta-RCNN is evaluated on animal subset of ImageNet-LOC. Animal subset of ImageNet-LOC only contains single animal category per image, so there are no irrelevant classes during training and it's simpler than the situation we discussed. In addition to FRCN and LSTD, we also include another latest baseline **RepMet** (Schwartz et al., 2019), which replaces FC classification layers in FRCN with more careful design of PN layers, as well as much more stronger backbone architecture (DCN (Dai et al., 2017) and FPN (Lin et al., 2017a)). In this benchmark, we have 50 categories per task, so we attach a 1x1 convolution layer before class-specific feature map generation to reduce computation cost. We report the results in Tab. 4. In 50way-1shot and 50way-5shot, Meta-RCNN is better than other methods.

| Model | Backbone | 50way-1shot | 50way-5shot |
|---|---|---|---|
| vanilla FRCN (Ren et al., 2015) | VGG16 | 16.5% | 34.3% |
| LSTD (Chen et al., 2018) | VGG16 | 19.2% | 37.4% |
| RepMet (Schwartz et al., 2019) | DCN+FPN | 24.1% | 39.6% |
| Meta-RCNN (ours) | VGG16 | 24.6% | 40.1% |
| Meta-RCNN (ours) | ResNet50 | **25.1%** | **40.3%** |

Table 4: mAP performance evaluation on IMAGENET-FSOD BENCHMARK.

### 5.4 DISCUSSIONS

*Extension to other Meta-Learning Methods:* Beyond prototypical networks, other meta-learning methods such as MAML (Finn et al., 2017) in principle can also be applied, e.g., we can apply MAML for vanilla FRCN framework, which updates the base model with the average gradient step of multiple tasks. However, in our experiments, the training process of MAML was unstable. This may be because few-shot detection is generally more difficult than few-shot classification, due to multiple dependent loss objectives (FRCN relies on RPN and regression loss etc.) and more complicated noisy contexts. In future, we plan to explore extensions to other meta-learning methods.

## 6 CONCLUSION

Object detection has been widely explored but little attention has been given to learning detectors under a few-shot regime. In this paper we propose a meta learning based detection algorithm Meta-RCNN, which is robust to few-shot learning, and the proposed training strategies make it more suitable in detection scenario. Specifically it adapts the Faster RCNN method and enables meta-learning of the object classifier, the RPN and the bounding box regressor. The RPN is meta-trained through a novel class-specific attention module. We conduct several experiments and obtain promising results.

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

## A  APPENDIX

### A.1  CATEGORY SPLIT IN VOC-FSOD BENCHMARK AND 2

Here we describe the category splits ($L_c$ and $S_c$) of VOC-FSOD benchmark and ImageNet-FSOD benchmark. These splits are used in all our paper.

**VOC-FSOD benchmark:**

$$L_c =$$

aeroplane, bicycle, 'bird, car, cat, chair, cow, person, pottedplant, tvmonitor

$$S_c =$$

bus, motorbike, train, dog, sheep, bottle, sofa, diningtable, horse, boat

**ImageNet-FSOD benchmark:**

$$L_c =$$

kit fox, English setter, Siberian husky, Australian terrier, English springer, grey whale, lesser panda, Egyptian cat, ibex, Persian cat, cougar, gazelle, porcupine, sea lion, malamute, badger, Great Dane, Walker hound, Welsh springer spaniel, whippet, Scottish deerhound, killer whale, mink, African elephant, Weimaraner, soft-coated wheaten terrier, Dandie Dinmont, red wolf, Old English sheepdog, jaguar, otterhound, bloodhound, Airedale, hyena, meerkat, giant schnauzer, titi, three-toed sloth, sorrel, black-footed ferret, dalmatian, black-and-tan coonhound, papillon, skunk, polecat, Staffordshire bullterrier, Mexican hairless, Bouvier des Flandres, weasel, miniature poodle, malinois, bighorn, fox squirrel, colobus, tiger cat, Lhasa, impala, coyote, Yorkshire terrier, Newfoundland, brown bear, red fox, Norwegian elkhound, Rottweiler, hartebeest, Saluki, grey fox, schipperke, Pekinese, Brabancon griffon, West Highland white terrier, Sealyham terrier, guenon, mongoose, indri, tiger, Irish wolfhound, wild boar, EntleBucher, zebra, ram, French bulldog, orangutan, basenji, leopard, Bernese mountain dog, Maltese dog, Norfolk terrier toy terrier vizsla, cairn, squirrel monkey, groenendael, clumber, Siamese cat, chimpanzee, komondor, Afghan hound, Japanese spaniel, proboscis monkey, guinea pig

$$S_c =$$

Pomeranian, wombat, hare, snow leopard, Arctic fox, Sussex spaniel, lynx, wood rabbit, Saint Bernard, redbone, chow, collie, German shepherd, affenpinscher, dingo, golden retriever, American Staffordshire terrier, briard, kelpie, Tibetan terrier, cocker spaniel, sloth bear, standard poodle, wire-haired fox terrier, Border terrier, American black bear, Bedlington terrier, banded gecko, wallaby, Tibetan mastiff, flat-coated retriever, koala, toy poodle, Border collie, Chesapeake Bay retriever, German short-haired pointer, great grey owl, Doberman, Lakeland terrier, miniature pinscher, timber wolf, hog, marmot, Irish setter, bull mastiff, Irish terrier, Shetland sheepdog, keeshond, miniature schnauzer, llama, Pembroke, ice bear, standard schnauzer, white wolf, Boston bull, Gordon setter, Great Pyrenees, Irish water spaniel, warthog, Scotch terrier, Chihuahua, Norwich terrier, Rhodesian ridgeback, borzoi, gibbon, Samoyed, tabby, Kerry blue terrier, Labrador retriever, thunder snake, Ibizan hound, beagle, curly-coated retriever, African hunting dog, boxer, common newt,

giant panda, ringneck snake, Angora, beaver, lion, bluetick, basset, alligator lizard, armadillo, pug, Greater Swiss Mountain dog, hognose snake, dhole, echidna, sidewinder, Komodo dragon, silky terrier, Brittany spaniel, patas, European fire salamander, Madagascar cat, macaque, boa constrictor, gorilla, polecat, howler monkey, Appenzeller, Blenheim spaniel, Indian cobra, Shih-Tzu, baboon, kuvasz, horned viper, rhinoceros beetle, tailed frog, Eskimo dog, Gila monster, mud turtle, capuchin, spider monkey, Leonberg, garter snake, African chameleon, barracouta, bullfrog, spotted salamander, leatherback turtle, rock python, marmoset, otter, Arabian camel, gar, tarantula, langur, tench, platypus, Italian greyhound, box turtle, cheetah, hippopotamus, English foxhound, eft, admiral, night snake, whiptail, siamang, agama, bittern, terrapin, axolotl, African grey, African crocodile, frilled lizard, quail, water ouzel, sulphur-crested cockatoo, bison, bustard, bulbul, cock, prairie chicken, ruffed grouse, jay, partridge, tusker, spoonbill, green snake, junco, black grouse, crane, water buffalo, toucan, redshank, hornbill, ostrich, vine snake, hummingbird, Indian elephant, magpie, albatross, king snake, little blue heron, bald eagle, peacock, limpkin, hamster, ruddy turnstone, jacamar, green mamba, kite, indigo bunting, American egret, American coot, coucal, house finch, ptarmigan, black stork, robin, white stork, brambling, red-backed sandpiper, king penguin, goldfinch, lorikeet, water snake, macaw, drake, vulture, bee eater, hen, dowitcher, red-breasted merganser, ox, diamondback, oystercatcher, goose, pelican, black swan,

