# OpenReview forum: "Meta-RCNN: Meta Learning for Few-Shot Object Detection"
_ICLR.cc/2020/Conference — Reject_

### Official Review · AnonReviewer1 · 2019-10-19
**Official Blind Review #1**

**Rating:** 6

**Review:**

In this paper, authors propose a meta-learning based approach for low-shot object detection. Specifically, they use prototype in the support set as attention guidance, and learn the category-specific representation for each query image. Subsequently, they use the style of Faster RCNN for object detection.

It is an OK paper with good structure. The idea is somewhat novel, in terms of meta-learning based low-shot detection framework. My main concern is about experiment. First, the data setting is branch new. Why not use the data setting in the literature, e.g., COCO to VOC in LSTD (Chen et al., 2018)? As a result, how to make a fair comparison bothers me a little. Furthermore, LSTD is a non-episodic approach. How to make it in a meta-learning way? Please clarify the implementation details for all other related works in the comparison.

**Experience Assessment:**

I have published one or two papers in this area.

**Review Assessment: Checking Correctness Of Derivations And Theory:**

I assessed the sensibility of the derivations and theory.

**Review Assessment: Checking Correctness Of Experiments:**

I assessed the sensibility of the experiments.

**Review Assessment: Thoroughness In Paper Reading:**

I read the paper at least twice and used my best judgement in assessing the paper.

---

> ### Author Response · Authors · 2019-11-15
> **We offer clarifications on why the experiment setting is fair**
>
> Thanks for the comments! We agree with your concerns, and would like to offer clarifications for a clearer understanding.
>
> To do a novel few-shot detection task, a prior needs to be acquired from some base data (e.g. meta train data in our case). To acquire this prior, we can follow two approaches: 1) Train a traditional model (e.g. a detector or classifier), and then fine tune on the novel few-shot task; OR 2) Acquire a prior via meta-learning on the base data, and learn a model that is trained to do few-shot learning.
>
> LSTD follows the first paradigm, while our proposed Meta-RCNN follows the second paradigm. Note that both methods have access to the exact same base data, i.e., they have access to the same information. They differ only in the learning algorithm. Then, a novel few-shot task is given to the algorithm, and the algorithm makes the prediction.
>
> Since both models have access to the same information, and make predictions on the same few-shot test task, the comparison is fair.
>
> Data Split Difference
> Meta-learning literature (Vinyals et al. 2016, Finn et al. 2017, Snell et al. 2017, etc.) evaluates few-shot performance over multiple tasks drawn from a test task distribution, i.e., the few-shot performance is measured and averaged over multiple tasks. This is a more reliable metric than evaluating performance on only one few-shot task. LSTD data split considers evaluation on only one few-shot task in their data split. We train LSTD on appropriate base data, and then evaluate its performance over multiple tasks, and compare this performance with our method.

---

### Official Review · AnonReviewer3 · 2019-10-22
**Official Blind Review #3**

**Rating:** 3

**Review:**

The paper proposes a method for few-shot object detection (FSOD), a variant of few-shot learning (FSL) where using a support set of few training images for novel categories (usually 1 or 5) not only the correct category labels are predicted on the query images, but also the object instances from the novel categories are localized and their bounding boxes are predicted. The method proposes a network architecture where the sliding window features that enter the RPN are first attenuated using support classes prototypes discovered using (a different?) RPN and found as matching to the few provided box annotations on the support images. The attenuation is by channel wise multiplication of the feature map and concatenation of the resulting feature maps (one per support class). After the RPN, ROI-pooling is applied on the concatenated feature map that is reduced using 1x1 convolution and original feature map (before attenuation) being added to the result. Following this a two FC layer classifier is fine-tuned on the support data to form the final
RCNN head of the few-shot detector. The whole network is claimed to be meta-trained end to end following COCO or ImageNet (LOC? DET?) pre-training. The method is tested on a split of PASCAL VOC07 into two sets of 10 categories, one for meta-training and the other for meta-testing. In addition, experiments are carried out on ImageNet-LOC animals subset. In both cases, the result are compared to some baselines, and some prior work.

Although FSOD is an important emerging problem, and advances on it are very important, I believe there are still certain gaps in the current paper that need to be fixed before it is accepted. Specifically:

1. Some important details are missing from the description. For example, detectors are usually trained on high resolution images (e.g. 1000 x 1000) and hence are problematic to train with large batches, yet in the proposed approach it is claimed that the proposed model is meta-trained with batch size 5 on 5 way tasks with 10 queries each, so even in 1-shot case, does it mean that 5 x 15 = 75 high resolution images enter the GPU at each batch? I doubt that even in parallel mode with 5 GPUs and 15 high res image per GPU it is possible for claimed backbone architectures (ResNet-50 and VGG16).
As another example, the details of fine-tuning during meta-training seem to be left out, is the model optimized with an inner loop? Details of the RPN that is used to select the support categories prototypes are not specified, where it comes from and how is it trained (clearly as the "main" RPN relies on attenuated features, it cannot be it)? Some additional technical details are not very clear and hinder the reproducibility of the paper (no code seem to be promised?), in general I suggest the authors to improve the writing and clarity of the paper.

2. In VOC07 experiment, FRCN-PN is very vaguely described and being claimed that it stands for RepMet (Karlinksy et al., CVPR 2019). It is not clear what it is and its training procedure on VOC07 is not clearly described.
It is also claimed in ImageNet experiment that the real RepMet is "more carefully designed then FRCN-PN" and has a better backbone, hence it is not clear why FRCN-PN should stand for it.
I suggest the authors to either do a direct comparison or remove their claim of comparison.

3. RepMet paper has proposed an additional benchmark on ImageNet-LOC with 5-way 1/5/10-shot episodes, and afaik it is reproducible as its code is released, so I am wondering as to why it was not used for
evaluation given that the authors made the effort of reproducing another ImageNet-LOC test on the same categories? It should be evaluated for a fair comparison.

4. Although they don't strictly have to compare to it, I am wondering if the authors would be willing to relate to a similar approach that was proposed for the upcoming ICCV 19:
"Meta R-CNN : Towards General Solver for Instance-level Low-shot Learning", by Yan et al. Their approach is more similar to RepMet in a sense that the meta-learning is done in the classifier head,
and better results are reported on VOC07 benchmark (and except for 1-shot, higher results are reported for the 3 and 5 shot FRCNN fine-tuning).

**Experience Assessment:**

I have published one or two papers in this area.

**Review Assessment: Checking Correctness Of Derivations And Theory:**

I assessed the sensibility of the derivations and theory.

**Review Assessment: Checking Correctness Of Experiments:**

I carefully checked the experiments.

**Review Assessment: Thoroughness In Paper Reading:**

I read the paper thoroughly.

---

> ### Author Response · Authors · 2019-11-15
> **Thanks for highlighting potential issues! We think they can be addressed.**
>
> Thanks for the comments! We do agree with some of your concerns, but do think that most of the suggested issues are addressable.
>
> 1. Implementation details with high-res images
>
> Thanks for the suggestions, and we apologise for lack details.
>
> During meta-training, we train the model using 5-way-1shot tasks, and only 5 query images (1 query image per class). This results in a total of 10 images for one task. With this, implementing the meta-training is not too difficult. Using this trained model, we evaluate performance on various settings (e.g. 5-way 1-shot, and 5-way 5-shot) meta test tasks. We apologize for the lack of clarity in the first RPN - this is not a critical component, and just a minor trick we use to make the prototype more robust (instead of constructing prototypes of the support objects by directly using ground truth bounding boxes and labels, we also use the proposals generated by the RPN, and if it has sufficient overlap with the ground truth, it is used for constructing the prototype). The main contribution in RPN is the one that is trained to generate support(class)-specific proposals.
>
> We will definitely release the code.
>
>
> 2 & 3. Regarding RepMet
>
> Thanks for these comments regarding RepMet.
> i) We have improved the presentation to not call it RepMet, but to call it FRCN-PN(baseline), and have changed the written section describing the relation of FRCN-PN with RepMet.
> ii) FRCN-PN shares a similar principle as RepMet (traditional detector training + replacing the object classifier with a meta-learner), and thus is a baseline we considered for our work.
> iii) We would have liked to reproduce RepMet and compare directly with the original method, however, we were not able to find the code for it. As a result, we decided to implement the method based on this principle ourselves as a baseline.
> iv) The code for RepMet: The arxiv version and the published version do not have a working link for the code being available online. We found a not well tested/incomplete version of the code  ( https://github.com/HaydenFaulkner/pytorch.repmet ) done by a third party, which has not yet reproduced the results.
>
>
> 4. Comparison with Meta-RCNN in ICCV2019
>
> Thanks for suggesting the reference. We believe this work was done in parallel with our work. We would like to highlight that this work was made available on arxiv (28th September) a few days after the ICLR submission deadline (25th September). Moreover, it appeared in ICCV even more recently (27th October).
>
> This work does share some similarities as our work (principle of class-attentive module), however, there is a fundamental difference in the training approach, specifically for the RPN. In contrast to the reference paper, our RPN is meta-trained and is tailored to generate proposals for the few-shot setting.
>
> We train the RPN in the meta-learning paradigm (meta-RPN), whereas the RPN training in the ICCV paper is trained using the traditional setting. This difference is extremely crucial for few-shot detection. Traditional RPNs will detect all objects in the image (including objects not of interest, i.e., they will even detect objects that are not available in given support set). Our meta-trained RPN generates proposals for an object from classes only belonging to the support set (i.e., it generates class-specific proposals).
>
> Finally, we would also like to highlight that following the meta-learning literature, we have evaluated the performance of the object detector on “multiple” few-shot detection tasks. Our reported few-shot performance is average performance over these tasks, in contrast to the existing reference which evaluates result on exactly one few-shot task.
>
> As regards empirical comparisons, it would be slightly time consuming to do this given different settings (e.g. hyperparameters, backbone, data splits, different approach for using the meta-train dataset, etc.). We do aim to do this in the future.

---

### Official Review · AnonReviewer2 · 2019-10-27
**Official Blind Review #2**

**Rating:** 8

**Review:**

This paper is about the task of object detection in the setting of few-shots dataset. The problem is addressed in the learning scheme of meta-learning paradigm: the proposed meta-rcnn trains the popular faster-rcnn on several tasks of few shots object detection while the RPN and the object classification networks are meta-learned among the tasks. Compared to previous work the paper introduces the meta learning framework and several changes to the faster rcnn detector. A prototype representation is derived from the standard RPN network and its proposed bounding box. An attention mechanism choose the object of interest and is used to train the final RPN and classification network. Experiments on the popular Pascal Voc 2007 and ImageNet-FSOD show that the proposed system have state of the art performance.

The paper is very well written, easy to read and of excellent presentation. The introduction of the meta learning paradigm and its use to learn the RPN and classification networks are incremental in novelty but interesting. The experiments are solid and show state of the art performance. As a result I recommend this paper to be accepted.

Minor issues:
- in caption of Fig1: avialable -> available
- in 4.1: “Compared to other variants...” please add a reference to the specific methods you are comparing to.

**Experience Assessment:**

I have published in this field for several years.

**Review Assessment: Checking Correctness Of Derivations And Theory:**

I carefully checked the derivations and theory.

**Review Assessment: Checking Correctness Of Experiments:**

I carefully checked the experiments.

**Review Assessment: Thoroughness In Paper Reading:**

I read the paper at least twice and used my best judgement in assessing the paper.

---

> ### Author Response · Authors · 2019-11-15
> **Thanks for the positive comments and the concerns!**
>
> Thank you for your review! We were delighted with your comments!
>
> As regards novelty, we would like to highlight that it is not trivial to adapt meta-learning for object detection, and to the best of our knowledge, ours is the first work that trains both the object classifier and the RPN in a meta-learning paradigm, making all the components tailored for  few-shot detection.
>
> Thanks for identifying the writing issues, we have fixed them in the current version.

---

### Decision · Program_Chairs · 2019-12-19

**Decision:**

Reject

**Comment:**

This paper develops a meta-learning approach for few-shot object detection. This paper is borderline and the reviewers are split. The problem is important, albeit somewhat specific to computer vision applications. The main concerns were that it was lacking a head-to-head comparison to RepMet and that it was missing important details (e.g. the image resolution was not clarified, nor was the paper updated to include the details). The authors suggested that the RepMet code was not available, but I was able to find the official code for RepMet via a simple Google search:
https://github.com/jshtok/RepMet
Reviewers also brought up concerns about an ICCV 2019 paper, though this should be considered as concurrent work, as it was not publicly available at the time of submission.
Overall, I think the paper is borderline. Given that many meta-learning papers compare on rather synthetic benchmarks, the study of a more realistic problem setting is refreshing. That said, it's unclear if the insights from this paper would transfer to other machine learning problem settings of interest to the ICLR community.
With all of this in mind, the paper is slightly below the bar for acceptance at ICLR.